# Development of an Electrostatic Beat Module for Various Tactile Sensations in Touch Screen Devices

**Young-Bok Joo [1,†], Eun-Jae Shin [1,†], Yong Hae Heo [1], Won-Hyeong Park [1], Tae-Heon Yang [2] and Sang-Youn Kim [1,*]**

[1] Interaction Laboratory, Advanced Research Technology Center, Computer Science and Engineering, Korea University of Technology and Education, Chungjeol-ro, Byeongcheon-myeon, Dongnam-gu, Cheonan, ChungNam 330-708, Korea; ybjoo@koreatech.ac.kr (Y.-B.J.); ejshin@koreatech.ac.kr (E.-J.S.); huice@koreatech.ac.kr (Y.H.H.); ipo1001@koreatech.ac.kr (W.-H.P.)

[2] Department of Electronic Engineering, Korea National University of Transportation, 50, Daehak-ro, Daesowon-myeon, Chungju-si, Chungcheongbuk-do 57615, Korea; thyang@ut.ac.kr

\* Correspondence: sykim@koreatech.ac.kr; Tel.: +82-41-560-1461

† These authors contributed equally to this work.



**Featured Application:** This paper is an extended version of our paper published in Proceedings of the International AsiaHaptics Conference 2016, Kashiwanoha, Japan, 29 November–1 December 2016; pp. 181–184. In this paper, we presented a detailed design of the electrostatic beat (ETB) module, its fabrication process and its operating principle. Furthermore, we quantitatively and qualitatively investigated the vibrational behavior of the proposed ETB module.

**Abstract:** One of the most dominant factors in developing tactile modules is the ability to generate abundant vibrotactile sensation. This paper presents a new vibrotactile module which can stimulate two mechanoreceptors at the same time without any mechanical vibration motors. To realize that, we first design an electro-tactile beat module (an ETB module) consisting of a lower part, a connection part and an upper part. The two electrodes were designed in an interdigitated pattern and were applied to the upper part. By applying two voltage inputs with slightly different frequencies to two electrodes in the proposed ETB module, respectively, we can create beat-patterned vibration. Furthermore, we can create normal vibration with the proposed ETB module by applying same frequency to the two electrodes. Experiments were conducted to validate the haptic performance of the proposed prototype. The results show that the proposed ETB module can create not only beat-patterned vibration but also normal vibration. The results also show that it can generate strong enough vibration to stimulate mechanoreceptors in wide frequency ranges.

**Keywords:** tactile display; haptics; beat; vibrotactile; electric beat; electro tactile beat

## 1. Introduction

Various types of mobile consumer-electronic devices based on a touch screen panel (TSP), for instance a navigation, a smart phone and a tablet personal computer (PC), have been spread out widely. In interacting with mobile consumer-electronic devices, visual information is the most important factor because human eye is one of the most highly developed organs and its actions are well documented. Hence, it is important to transfer a large amount of visual information to a user for immersive interaction with consumer electronics.

While visual information is the most dominant sensory input for perceiving an object, haptic information coupled with visual information is also important factor for intuitive and immersive

interaction between human and consumer electronic devices. The reason is that touch-based interaction is the first and most natural way and it allows a user to non-verbally and cognitively interact with devices [1]. Haptic feeling consists of kinesthetic sensation (sensory data obtained by receptors in joints, muscles and ligaments) and tactile sensation (sensory information acquired by pressure receptors in the skin). Many kinesthetic actuators are too bulky to be inserted into mobile consumer electronic devices. Therefore, for creating haptic feeling in small-sized mobile consumer electronic devices, many vibrotactile actuators, which can be easily fabricated in small size, have been developed [2–5] and some of them were successfully commercialized [6,7]. D. Pyo et al. developed an impact type of vibrotactile actuator which has wide operating frequency range for mobile applications [2]. I. Poupyrev et al. presented a thin vibrotactile actuator by applying adhesive electrodes onto the both side of thin piezoceramic film [3]. F. Pece et al. suggested a flexible and wearable haptic module based on magnetically actuated motor for localized haptic and tactile feedback [4]. W.H. Park et al. presented a soft vibrotactile actuator based on an electroactive polyvinyl chloride (PVC) gel [5]. Based on these actuators, many tactile rendering methods how to generate abundant haptic sensation have been suggested. J. Van Erp et al. suggested a method how to construct a vibrotactile display system using vibrotactile actuators [8]. A. Chang et al. developed a mobile platform and a method for multimodal communication between users [9]. J. Linjama et al. addressed a gesture-haptic interaction method using an accelerometer and a vibration motor [10]. I. Oakley et al. developed a hardware platform with the ability to convert the sensed motion into vibrotactile output during scrolling task [11,12]. Y. Sekiguchi et al. developed a mobile platform to generate clattering sensation during shaking using a solenoid-based vibrotactile actuator [13]. T. Kaaresoja and J. Linjama found out that the duration of a control signal for generating vibration should be between 50 and 200 m sec [14]. These tactile actuators and tactile rendering methods stimulate human mechanoreceptors in skin to create vibrotactile sensation.

In glabrous skin, there are four major mechanoreceptors (Meissner corpuscle, Merkel's disk, Ruffini ending and Pacinian corpuscle) [15]. Merkel's disk responds to quasi-static deformations of the skin, such as force or displacement in the frequency ranging from 0.4 Hz to 3 Hz. It plays an important role in detecting spatial structures in static contact, such as an edge or a bar. Ruffini ending produces a buzz-like sensation in the frequency ranging between 100 to 500 Hz. Meissner corpuscle with a frequency ranging from 2 Hz to 40 Hz, detects the dynamic deformations of the skin such as the sensation of flutter. Pacinian corpuscle, which has a frequency response in the range of 40 Hz to 500 Hz, is the most sensitive to vibration amplitude and is particularly known to serve as the detector of acceleration or vibration. Currently, consumer electronic devices provide many virtual objects having complex shape or heterogeneous surface. Currently, many virtual objects have been used in consumer electronic devices. Users who interact with these objects want to haptically sense the presence of the target objects and their surface properties such as texture and roughness. For satisfying users' demand, two or more mechanoreceptors need to be stimulated at the same time when he/she interacts with various surfaces.

Beat vibration can be a solution for stimulating two mechanoreceptors at the same time. The beat vibration is an interference pattern of two different vibrations having slightly different frequencies [16]. The beat vibration has two different frequencies so that it can stimulate two mechanoreceptors at the same time. Because of this feature, the beat vibration allows users to perceive special touch sensation unlike normal vibration. Let us consider the case where two haptic actuators, which are actuated at the frequency of $f_1$ and $f_2$, respectively, are attached to a mobile device. If two actuators have slightly different operating frequency, the mobile device vibrates at the frequency of $(f_1 + f_2)/2$. At this time, the envelope frequency of the vibration (we call it beat frequency) becomes $f_1 - f_2$. If $f_1$ is 206 Hz and $f_2$ is 200 Hz, the mobile device vibrates at 203 Hz and its beat frequency is 6 Hz. In this case, Meissner corpuscle responses to 6 Hz vibration and Pacinian corpuscle is stimulated to vibration at 203 Hz. Therefore, vibration signal with beat can stimulate two mechanoreceptors (Meissner corpuscle and Pacinian corpuscle) at the same time.

There have been a few research works for creating beat vibration [17–21]. S.C. Lim et al. developed a pin-array tactile device and generated haptic beats by providing two pins with slightly different sinusoidal operation frequencies [17]. S. Yang et al. created haptic beats using two vibrotactile actuators and investigated users' ability to distinguish emergent haptic beats from pairs of pure vibration presentations as a function of body location and frequency differences [18]. P. Tranchant et al. developed a tactile module for generating beat and they investigated beat synchronization to vibrotactile electronic dance music in hearing and deaf people [19]. M. Konyo et al. suggested a tactile synthesis method using multiple frequency vibrations [20]. Makino and Maeno investigated the perception of haptic beats when a user touched a mobile screen by actuators [21].

To create the beat vibration in their work, two or more vibrotactile actuators have to be inserted into the mobile device. Due to the two or more vibration motors, the conventional beat vibration module needs additional power consumption and additional cost. Furthermore, the beat module requires additional space to mount the two or more actuators. Another problem in beat vibration modules using vibrotactile actuators is that a conventional vibration module has narrow frequency bandwidth. To simulate the various virtual objects with the beat vibration, a new way with no or small number of actuators is demanded.

In this paper, we present a new electro tactile beat (ETB) module, which has wide operating frequency range and create beat vibration without any mechanical vibrotactile actuators. The proposed ETB module is composed of three parts: a lower part, a connection part and an upper part. The upper part is connected to the lower part via the connection part (four connectors). We design interdigitated pattern on the upper side of the lower part using two electrodes. These two electrodes are used for an anode and there is an electrode in the upper part for a cathode. When two voltage inputs with slightly different frequencies are applied to the interdigitated electrode pattern, two electric fields are generated between the interdigitated pattern and the cathode. At that time, the created electric fields influence each other and then finally they are transformed to a new electric field having beat pattern in the lower part. The beat-patterned electric field makes electrostatic attraction force between the upper part and the lower part and the created electrostatic attraction force causes the lower part to vibrate. This vibration is propagated to the upper part through the connection part, so a user can feel beat vibration whenever he/she touches the upper part. We define the vibrations generated by interfering two different electric fields as electric beat vibration. The frequency of the electric beat vibration can be easily changed by adjusting the frequencies of the input voltages. It means that the proposed ETB module can simulate a lot of virtual surfaces. We investigate the vibration behavior according to the electric beat vibration through experiments. The results show that the proposed ETB module can generate the beat vibration without mechanical vibrotactile actuators and can create strong vibrations enough to stimulate user's fingertip.

## 2. Design of the Proposed ETB Module

Figure 1a shows a schematic illustration of the proposed electro tactile beat (ETB) module consisting of an upper part, a connection part and a lower part. The connection part consisting of four connectors having 0.5 mm height connects the upper part to the lower part. Figure 1b shows the fabricated ETB module with dimension of 100 mm (H) × 200 mm (W) × 3.5 mm (T). Figure 2 shows the fabrication process of the upper part. An acrylic plate was prepared for making the upper part and it was carved and drilled using a layer cutter for assembling with the lower part. After that, silver nanowire was sprayed (thickness: 0.1 µm) onto the bottom side of the plate to make an electrode (we call this electrode as an electrode 1). Finally, we could obtain the upper plate (whose thickness is 2 mm) with an electrode.

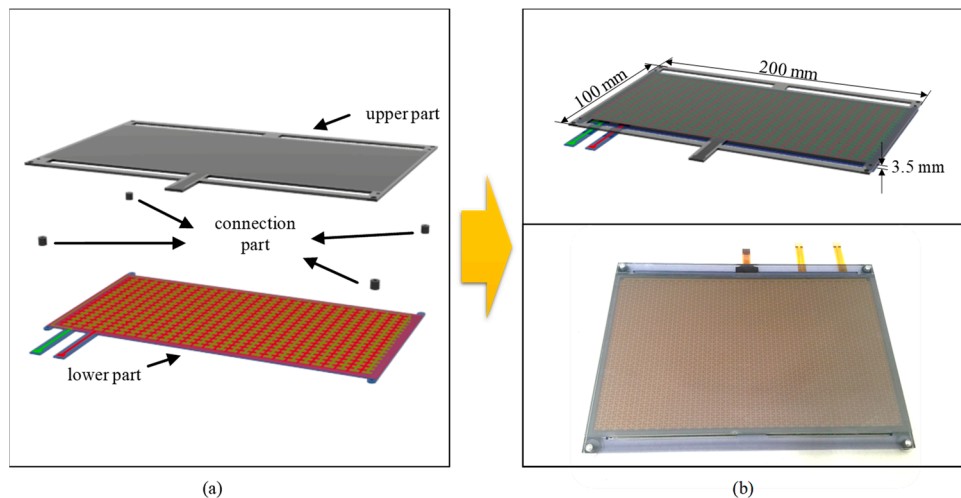

**Figure 1.** Fabricated electro tactile beat (ETB) module consisting of an upper part, a connection part and a lower part, (**a**) its overall structure, (**b**) the fabricated ETB module.

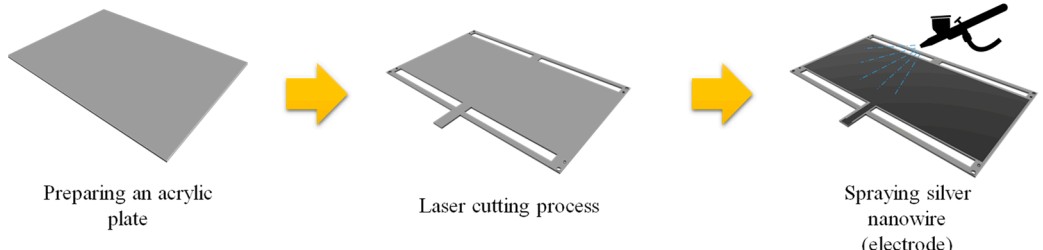

**Figure 2.** Fabricated process of the upper part.

Figure 3a shows the structure of the lower part consisting of three components; a dielectric layer, an active layer (electrodes 2 and 3) and a bottom plate layer. The dielectric layer (thickness: 100 μm) and the bottom layer (thickness: 1 mm) are made of a polyimide film and a glass epoxy laminate, respectively. The dielectric layer is used for preventing electric short circuit between the electrodes 2 and 3. The active layer is sandwiched between the dielectric layer and the bottom layer. Two voltage inputs are applied to the electrodes 2 (green) and 3 (red). The electrodes 2 and 3 take the form of interdigitated pattern with a gap of 0.2 mm. Figure 3b shows the fabrication process of the active layer. An epoxy glass fiber plate was prepared and an aluminum mask was also prepared for sputtering electrodes onto the epoxy glass fiber plate. We put the aluminum mask on the epoxy glass fiber plate and then we sputtered copper on the epoxy glass fiber plate and the mask. By removing the mask, we obtained the active layer with patterned electrodes (electrodes 2 and 3).

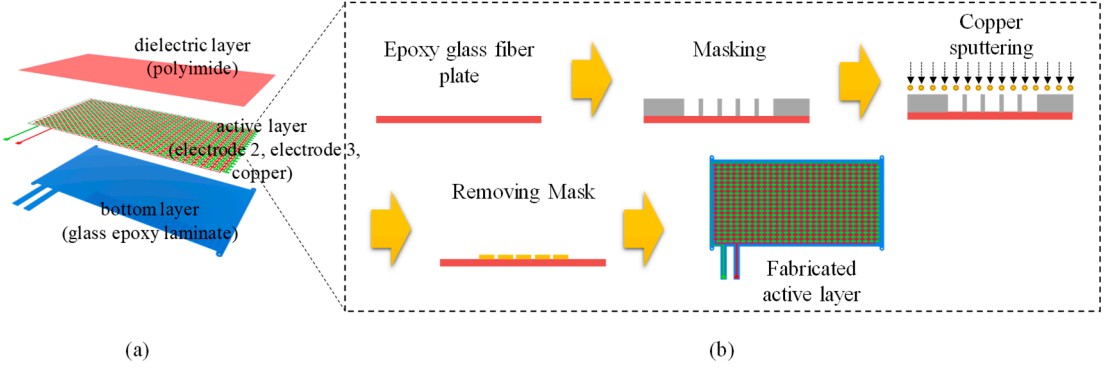

**Figure 3.** Structure of the lower part and fabrication process of the active layer, (**a**) component of the lower part, (**b**) fabrication process of the active layer.

　　　　Figure 4 shows the operating principle of the proposed ETB module. The electrode 1 in the upper part is used for a cathode and the other two electrodes (electrodes 2 and 3) in the lower part are used for anodes. Figure 4a shows the initial state (no voltage inputs) of the proposed ETB module. Let us consider a case where two alternating current (AC) sinusoidal input voltages with slightly different frequencies are applied to the electrodes 2 and 3, respectively as shown in Figure 4b. According to the applied input voltages, electrodes 2 and 3 generate two different electric fields. Since the two electrodes were covered by the dielectric layer and a gap between the electrodes 2 and 3 is too small, the two electric fields have direct influence on each other. That is, the two input voltages are combined to create new electric signals which make an interference voltage pattern as shown in Figure 4c. We define the new electric signal as electric beat. We can generate a variety of electric beats by adjusting the frequencies of the two input voltages. The electric beat makes electrostatic attraction force between the electrode 1 and the dielectric layer having electrodes 2 and 3 (Figure 4d) and the created electrostatic attraction force makes the upper plate and lower plate bend. If we provide zero voltage, the two layers are reshaped back to their original states. So, AC voltage input makes the two layers vibrate and finally a user can feel the vibration on the upper plate as shown in Figure 4e.

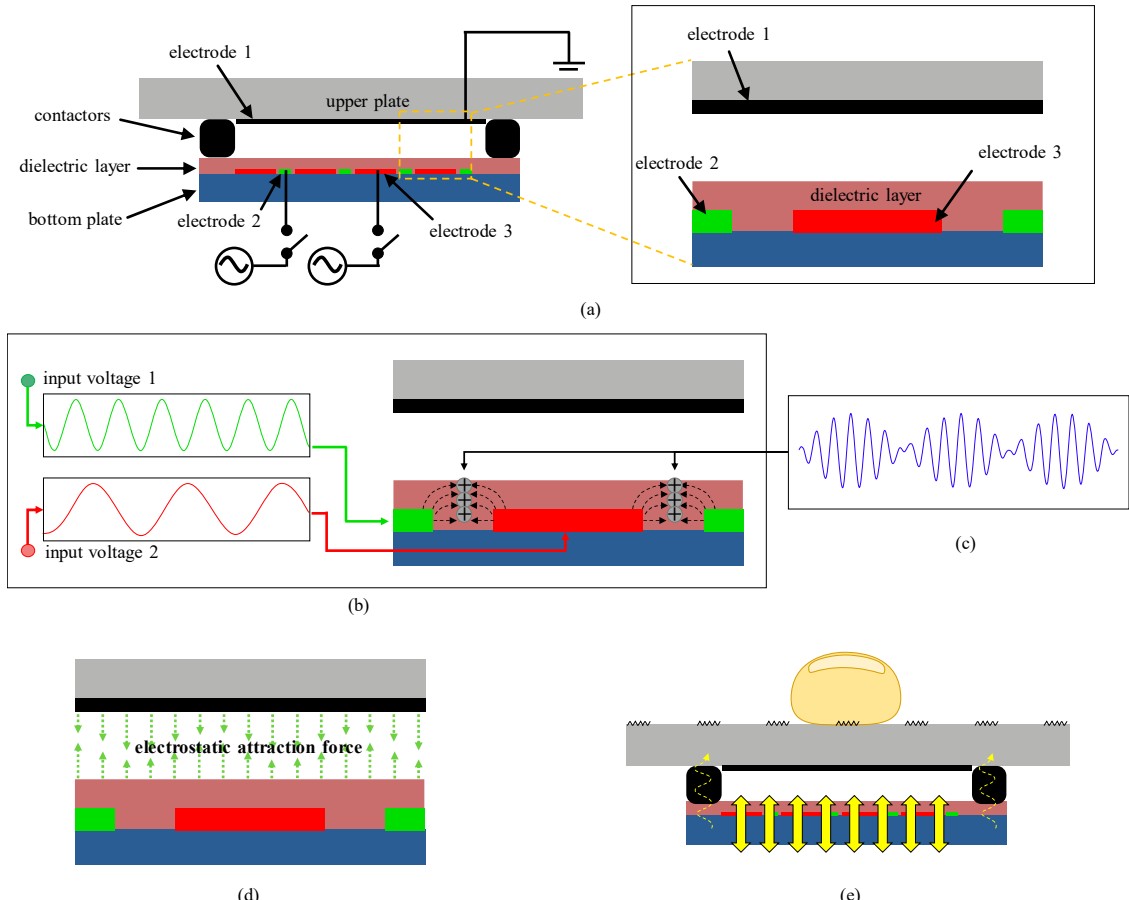

**Figure 4.** The operating principle of the ETB module. (**a**): initial state of the ETB module. (**b**): partial magnification of the ETB module. (**c**): An influence between two input voltages in the lower plate. (**d**): An interference pattern by the two input voltages. (**e**): A generated electrostatic attraction force between the upper and bottom plates.

　　　　We designed a portable ETB platform consisting of a control part, a circuit part and an operation part as shown in Figure 5a. Through the control part, we can adjust the two input frequencies which will be applied to the ETB module and furthermore we can watch the two frequencies. There are two knobs in the lower side of the control part to easily adjust the frequencies of the two input voltages.

The circuit part consists of a microprocessor (ATMega 128, Microchip Technology Inc., Chandler, AZ, USA), a high voltage amplifier module (IMAGIS, IMAGIS Co. Ltd., Suwon-si, Gyeonggi-do, Korea) and four opto-couplers (OC100HG, Voltage Multipliers Inc., Visalia, CA, USA) as shown in Figure 5b. The microprocessor analyzes the adjusted frequencies by rotating the two knobs in the control part and it creates AC input wave with the adjusted frequency. The input voltages are amplified by the high voltage amplifier module and then are applied to the ETB module through the four opto-couplers.

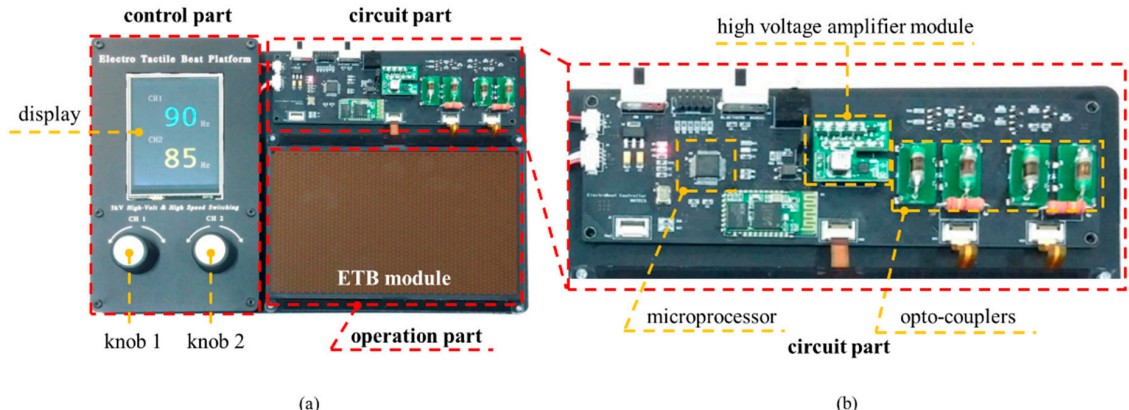

**Figure 5.** The designed ETB module. (**a**): Constructed portable ETB platform consisting of a control part, a circuit part and an operation part. (**b**): Magnified image of the circuit part.

## 3. Experiment and Result

As we mentioned before, two voltage input signals were used for creating electrotactile beat (ETB) in the proposed module. Let us consider a case where the frequency of a first voltage input signal is set as the resonant frequency and the frequency of another voltage input is almost same (not exactly same) as the resonant frequency. In this case, the vibration amplitude created from the proposed ETB module will dramatically increases. To investigate the resonant frequency of the proposed ETB module, we observed the acceleration behavior. During this experiment, one voltage input was applied to electrodes 2 and 3.

Figure 6 shows the experimental setup consisting of a function generator (Protek 9305, Protek, Incheon, Korea), an accelerometer (Charge Accelerometer type 4393, Bruel&Kjaer, Naerum, Denmark), a high voltage amplifier (Trek 10/40A-HS, TREK, New York, NY, USA) and an oscilloscope (MSO/DPO 2000, Tektronix, Beaverton, OR, USA). The function generator was connected to a high voltage amplifier whose gain is 1000. We attached a mass of 100 g on the proposed ETB module and put the accelerometer on the mass of 100 g. AC voltage input (sine wave) of 1 $V_{PP}$ was created by the function generator and then was amplified by the high voltage amplifier in order to be conveyed to the proposed ETB module. The vibration force, which was created by the proposed ETB module, was measured by the accelerometer and the measured vibration force was displayed on the oscilloscope. In this experiment, the vibration force was obtained as a function of input frequency in a range of 1 Hz to 250 Hz.

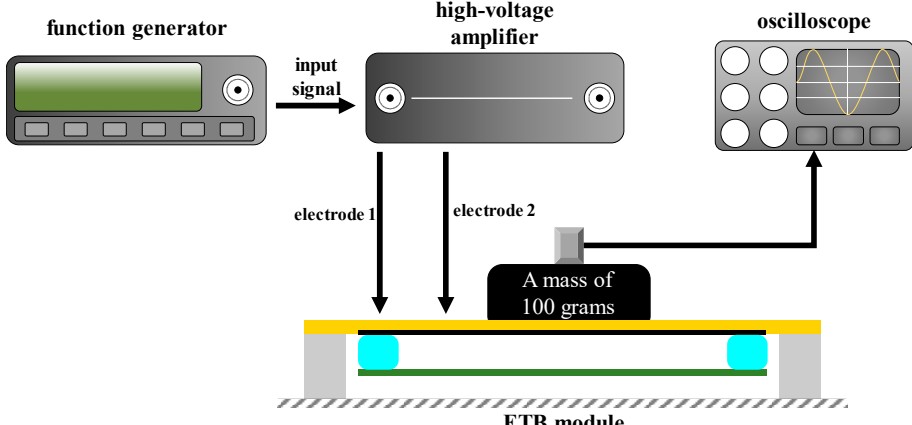

**Figure 6.** The experimental setup for measuring the resonant frequency of the proposed ETB module.

Figure 7 shows the acceleration behavior of the proposed ETB module as function of input frequency. The resonant peak in the proposed ETB module are shown at 90 Hz. The amplitude of the vibration is reached about 2.6 g (=9.8 m/s$^2$) at 90 Hz. As we mentioned before, the two voltage inputs are applied to the proposed module to create electric beats which will be used for stimulating human skin.

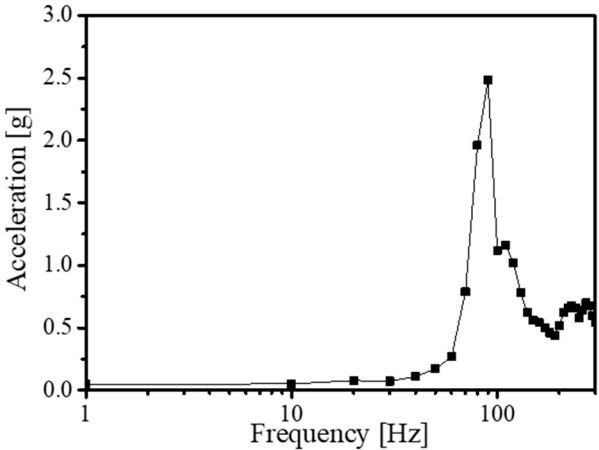

**Figure 7.** Vibrational behavior of the proposed ETB module as function of frequency.

To investigate the haptic behavior of the proposed ETB module, we constructed an experimental environment consisting of a frequency controller, two high voltage amplifiers (Trek 10/40A-HS, Trek, Lockport, NY, USA), an accelerometer (Charge Accelerometer type 4393, Bruel & Kjaer, Nærum, Denmark) and an oscilloscope (MSO/DPO 2000, Tektronix, Beaverton, OR, USA) as shown in Figure 8. In the experiment, the reference frequency was changed from 60 Hz to 150 Hz at 30 Hz interval (60 Hz, 90 Hz, 120 Hz and 150 Hz) and another frequency for generating the beat was slightly different from the reference frequency. This experiment was conducted with 12 frequency sets consisting of two AC sinusoidal input voltages (1kV) having slightly different frequencies, such as 60 Hz + 65 Hz, 60 Hz + 70 Hz, 60 Hz + 80 Hz, 90 Hz + 95 Hz, 90 Hz + 100 Hz, 90 Hz + 110 Hz, 120 Hz + 125 Hz, 120 Hz + 130 Hz, 120 Hz + 140 Hz, 150 Hz + 155 Hz, 150 Hz + 160 Hz, 150 Hz + 170 Hz. To create electrotactile beat, two voltage inputs having slightly different operating frequencies were generated the frequency controller. Two voltage inputs were applied to terminals 2 and 3 in the ETB module after being amplified by the high voltage amplifier and the ground signal was applied to the terminal 1. It is generally known that the electric field makes electric attraction force between two parallel electrodes and the amplitude of the electrostatic attraction force depends on the amplitude of the electric field. After attaching an

accelerometer (Charge Accelerometer type 4393, Bruel & Kjaer, Nærum, Denmark) onto the center of the contact plate of the ETB module, acceleration values are measured and recorded.

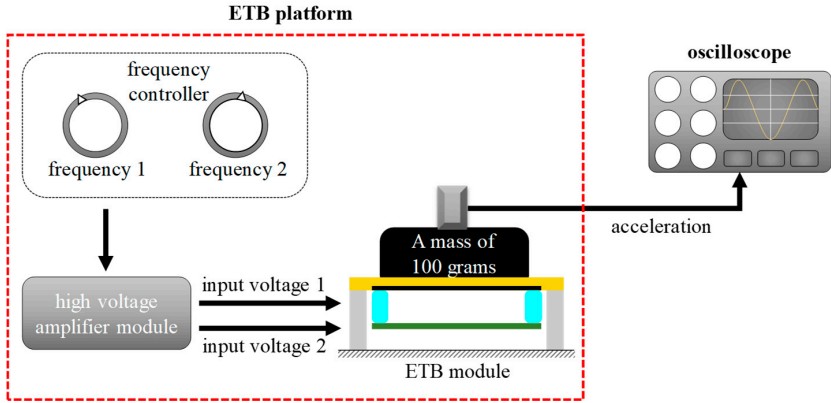

**Figure 8.** Experimental environment to investigate haptic behavior of the proposed ETB module.

Figure 9 shows the haptic behavior of the proposed ETB module as function of two input frequencies. The scale of the X-axis was also set as 1 s. As we expected, the results show that the measured acceleration signal has two frequencies which will be used for two mechanoreceptors at the same time. Let us take 90 Hz + 95 Hz as an example where we can easily find two vibration signals (one frequency is 5 Hz and the other is 92.5 Hz). It means that the proposed module can simulate not only Meissner corpuscle but also Pacinian corpuscle simultaneously. Also, the result shows the largest amplitude of vibration at a reference frequency of 90 Hz. In order to show that the proposed ETB module can create vibrotactile force sufficiently strong for transferring haptic information to users, we first measured the absolute vibrotactile threshold through a simple experiment and then compared the vibrational force obtained from the proposed ETB module and the absolute threshold. The experiment was conducted with 10 persons whose average age is 26.9 years. We depicted the absolute vibrotactile threshold (colored in red) in Figure 9. The vibrotactile strength obtained from the proposed ETB module is stronger than the absolute vibrotactile threshold. Therefore, acceleration created from the proposed ETB module is sufficiently strong for transferring haptic information to users.

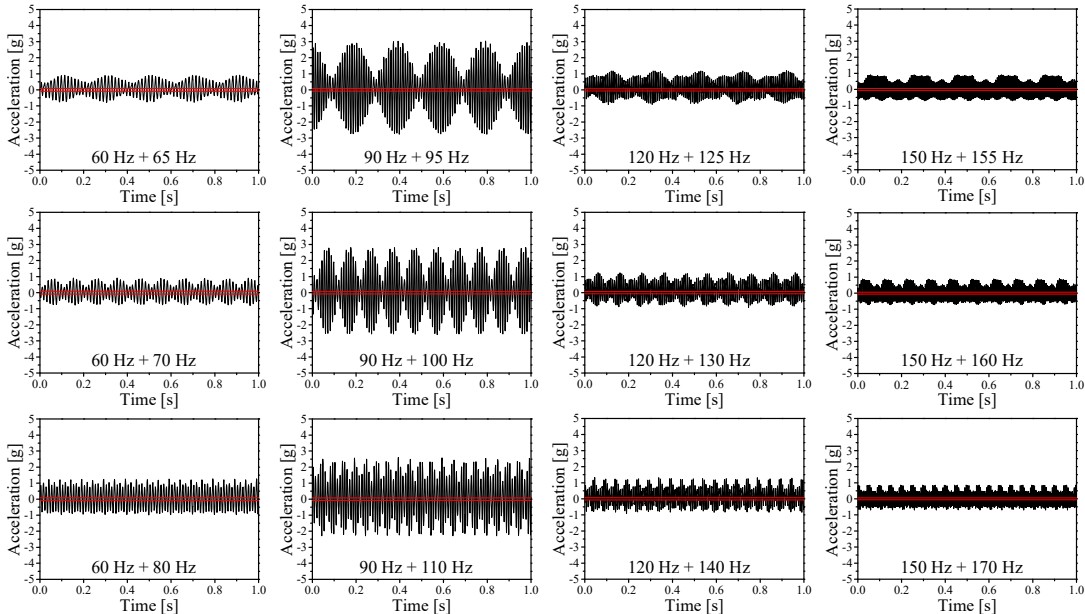

**Figure 9.** Vibration amplitudes of the ETB module.

In order to show that the proposed ETB module can create vibrotactile force sufficiently strong for transferring haptic information to users, we first measured the absolute vibrotactile threshold through a simple experiment and then compared the vibrational force obtained from the proposed ETB module and the absolute threshold. The experiment was conducted with 10 persons whose average age is 26.9 years. All of them were between 25 and 33 years old. Eight subjects were right-handed, and the others were left-handed, as determined by self-reports. All subjects do not have any sensorimotor abnormalities and they were paid after the experiment. Each subject was seated and instructed to touch the proposed ETB module with his/her index finger. After that, the subjects were presented with beat vibrations generated by the ETB module. We provided 12 vibration samples to each subject. Each sample is a beat vibration obtained from two input voltages having slightly different frequencies (60 Hz + 65 Hz, 60 Hz + 70 Hz, 60 Hz + 80 Hz, 90 Hz + 95 Hz, 90 Hz + 100 Hz, 90 Hz + 110 Hz, 120 Hz + 125 Hz, 120 Hz + 130 Hz, 120 Hz + 140 Hz, 150 Hz + 155 Hz, 150 Hz + 160 Hz, 150 Hz + 170 Hz). In each trial, three 1-s long vibrational stimuli were delivered to the user and there was a 250 ms pause between stimuli as shown in Figure 10. Among the three 1-s long vibration stimuli, one randomly selected interval contained a test stimulus and the other two contained no signal. After feeling the three 1-s long vibration stimuli, he/she found the test stimulus by pressing a corresponding keypad (1, 2 or 3). During this experiment, we set an amplitude of the initial stimulus as higher than a threshold which was found in pilot experiments and then we gradually reduced its amplitude. The subjects worn a headphone to remove sound noises from the proposed ETB module. Beep sound was conveyed to the subject via the headphone to inform the start and end of each trial to the subjects. We depicted the absolute vibrotactile threshold (colored in red) in Figure 9. The vibrotactile strength obtained from the proposed ETB module is stronger than the absolute vibrotactile threshold. Therefore, vibrational force created from the proposed ETB module is sufficiently strong for transferring haptic information to users.

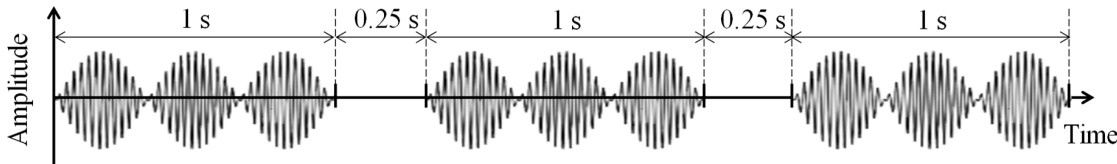

**Figure 10.** Experimental environment to measure the detection threshold.

We designed another experiment to show that users can discriminate vibrotactile beat and pure vibration (vibration without beat). An ABX test, which is a method of comparing the vibrotactile beat and the pure vibration, is used for this experiment. Ten subjects who joined in the previous experiment were participated in this test. Each subject was seated in front of the proposed ETB module and was instructed to touch the ETB module with his/her index finger. We prepared 12 reference vibrations (sample A, beat vibration, $f + (f + \Delta f)$) and we prepared three corresponding reference vibrations (sample B, pure vibration without beat, $f + f$, $(f + \Delta f /2) + (f + \Delta f /2)$, $(f + \Delta f) + (f + \Delta f)$) for each reference vibration. The subjects felt a known first reference vibration (sample A, beat vibration) during 1 s and took a rest for 1 s and then felt one of the known second corresponding reference vibrations (sample B, pure vibration without beat) for 1 s. After that, they were provided with unknown sample X (during 1 s) that is randomly selected from either sample A (beat vibration) or sample B (pure vibration). As soon as the subjects sense the sample X, they identify the sample X as either sample A or B. We tabulated the first reference vibration sets and corresponding reference vibration sets (Table 1).

**Table 1.** Reference vibration set for ABX test.

| Reference Vibration (A) | Corresponding Reference Vibration (B) | | |
| --- | --- | --- | --- |
| | 1st Corresponding Reference Vibration | 2nd Corresponding Reference Vibration | 3rd Corresponding Reference Vibration |
| 60 Hz + 65 Hz | 60 Hz + 60 Hz | 62.5 Hz + 62.5 Hz | 65 Hz + 65 Hz |
| 60 Hz + 70 Hz | 60 Hz + 60 Hz | 65 Hz + 65 Hz | 70 Hz + 70 Hz |
| 60 Hz + 80 Hz | 60 Hz + 60 Hz | 70 Hz + 70 Hz | 80 Hz + 80 Hz |
| 90 Hz + 95 Hz | 90 Hz + 90 Hz | 92.5 Hz + 92.5 Hz | 95 Hz + 95 Hz |
| 90 Hz + 100 Hz | 90 Hz + 90 Hz | 95 Hz + 95 Hz | 100 Hz + 100 Hz |
| 90 Hz + 110 Hz | 90 Hz + 90 Hz | 100 Hz + 100 Hz | 110 Hz + 110 Hz |
| 120 Hz +125 Hz | 120 Hz + 120 Hz | 122.5 Hz + 122.5 Hz | 125 Hz + 125 Hz |
| 120 Hz +130 Hz | 120 Hz + 120 Hz | 125 Hz + 125 Hz | 130 Hz + 130 Hz |
| 120 Hz +140 Hz | 120 Hz + 120 Hz | 130 Hz + 130 Hz | 140 Hz + 140 Hz |
| 150 Hz +155 Hz | 150 Hz + 150 Hz | 152.5 Hz + 152.5 Hz | 155 Hz + 155 Hz |
| 150 Hz +160 Hz | 150 Hz + 150 Hz | 155 Hz + 155 Hz | 160 Hz + 160 Hz |
| 150 Hz +170 Hz | 150 Hz + 150 Hz | 160 Hz + 160 Hz | 170 Hz + 170 Hz |

Each subject worn a headphone and was provided white sound noise to prevent the subject from taking a guess due to the vibration sound. Beep sound was conveyed to the subject via the headphone to inform the start and end of each trial to the subjects. To enhance the reliability, this procedure was repeated five times for each set. Because the number of subjects was 10, the total number of trials was 50. Figure 11 shows the result of ABX test. Interestingly, all subjects answered that the beat vibration is different from the pure vibration except three cases (60 Hz + 70 Hz, 60 Hz + 80 Hz and 120 Hz + 140 Hz). The Percentages of correct answers for (60 Hz + 70 Hz and 120 Hz + 140 Hz) are 96% and the percentage of correct answers for (60 Hz + 80 Hz) decreases to 88%. This is because two input frequencies are not close enough to create beat for above two cases. Therefore, we can conclude that the successfully created beat vibration is different from pure vibration (without beat).

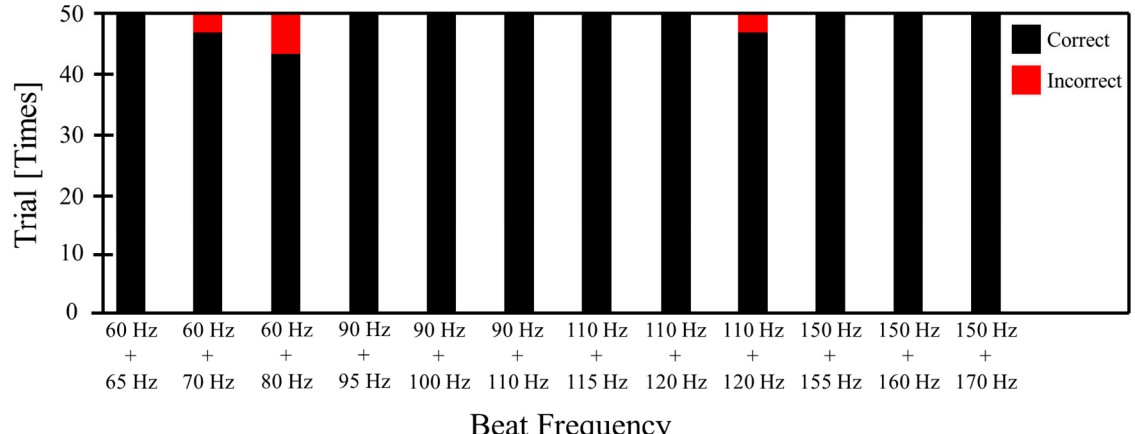

**Figure 11.** The result of ABX test.

## 4. Conclusions and Future Works

This paper suggested a haptic module, which does not need mechanical vibration motors, consisting of an upper part, a connection part and a lower part. Electric beat effect was used to simultaneously stimulate two mechanoreceptors in human's skin. To synthesize two electric fields, we designed interdigitated electrode pattern. When we applied two input signals having slightly different frequencies to the proposed module, electric beats were generated. The electric beats makes electrostatic attraction force between the upper part and the lower part and the electrostatic attraction force makes the lower part to vibrate. We verified that the proposed actuator could stimulate two human mechanoreceptors over a wide frequency range. We expected that the proposed ETB module

can be used for creating various tactile sensation of virtual objects in consumer electronic devices. Even though the proposed ETB module can convey more nuanced and more rich haptic sensation to users than pure vibration, the absence of vibrotactile rendering prevents the ETB module from simulating various surface. Therefore, we will study a vibrotactile rendering method which can simulate various surface including heterogeneous surfaces. The size of the electrode and the thickness of the acrylic plate are important factors for maximizing the performance of the actuator. Therefore, we are studying optimal the size of the electrode and the thickness of the acrylic plate to maximize the performance of the ETB module.

**Author Contributions:** Y.-B.J. and E.-J.S. designed the research problems; T.-H.Y. suggest idea and configuration the structure of the paper; E.-J.S., Y.H.H. and W.-H.P. conducted the experiments and analyzed the results; S.-Y.K. supervised the research. All authors discussed the results and wrote the paper.

**Funding:** This work was by Priority Research Centers Program through the National Research Foundation of Korea (NRF) funded by the Ministry of Education (NRF-2018R1A6A1A03025526). Also, this work was supported by the Technology Innovation Program (10077367, Development of a film-type transparent /stretchable 3D touch sensor /haptic actuator combined module and advanced UI/UX) funded by the Ministry of Trade, Industry & Energy (MOTIE, Korea).

**Conflicts of Interest:** The authors declare no conflict of interest.

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
