# Peer review of "Development of an Electrostatic Beat Module for Various Tactile Sensations in Touch Screen Devices"

_applsci, doi:10.3390/app9061229_

Round 1
Reviewer 1 Report
The authors proposed an electrostatic beat module composed of multiple electrodes to produce vibration at two frequencies. The module could successfully generate vibrations with beat as designed. However, the manuscript is not sufficiently well organized and not suitable for publication in the current form. Following are my comments:
1. Abstract is not well-organized and very redundant. Many sentences have very similar meaning.
2. Introduction does not sound logical. For example, the first paragraph claims the importance of visual information and then, suddenly, the second paragraph discusses the haptic information. The third one explains the anatomy of the finger skin and then the story jumps to the current electronic device.
3. Materials and fabrication of the device should be explained in the figure. They are written in the text part but hard to understand.
4. In Experiment and Result, the author claims the resonant frequency is important. But the authors claimed that the electrostatic form can change the actuation frequencies differently from the mechanical actuators (Line 69).
5. The results are shown in Figure 7. It would be better to FFT analyze them and show the beat.
6. The acceleration seems larger at ~90 Hz than the others due to the resonance. How can the device handle the difference in the real application?
7. Is the generated acceleration sufficiently strong for transferring haptic information? This paper does not show any haptic experiments and I cannot tell the effectiveness of this device.
Author Response
We would like to express our sincere appreciation for the reviewer’s comments. We have faithfully revised our paper (Title: Development of an Electrostatic Beat Module for Various Tactile Sensations in Touch Screen Devices, Article reference: applsci-435597) based on the reviewer's comments. Details of the revisions are appended in following sheets. Thank you very much in advance.

Reviewer 2 Report
This paper describes a vibration tactile display with an electrostatic actuator. The tactile display can provide beat vibration with two different electrostatic forces. The authors evaluated the frequency response. Also, the beat vibration was evaluated. I think that the hypothesis that the authors mentioned is unknown. Thus, I strongly recommend the authors to add experiment. I show the following points to improve the quality of the paper.
1. In the abstract, the authors used “;” instead of “,”. Is this typos?
2. In the abstract, the authors said “which can stimulate two or more human’s mechanoreceptors in real-time; without any vibrotactile actuators”. I think the proposed tactile display also provide the vibration with the electrostatic actuator.
3. The authors mentioned the effect of the beat vibration. However, I cannot agree with the effect. Can the beat vibration stimulate two different mechanoreceptors? If you have references about the effect, please add the references to show the effect. If not, please, reconsider the introduction.
4. Please explain the reason for the device design such as the thickness of acrylic plate and the size of the electrodes.
5. In the “3. Experiment and result”, the authors only show the experimental results. Please add detailed discussion. For example, the authors conduct FFT and confirm that the target frequencies are presented.
6. Where is the section 4?
7. Please conduct a sensory experiment. In the sensory experiment, the author should evaluate whether the provided beat vibration can provide different stimulus from normal vibration. I think that the experiment is indispensable to demonstrate the proposed concept. If the authors won’t conduct the sensory experiment, please reconsider the introduction.
8. The authors should add more references. There are many articles about tactile displays. At least, 20 or more references are required for the full length paper.
Author Response

(The authors gave the same response as above.)

Reviewer 3 Report
# Overview
The main contribution of the paper is an implementation of an electrostatic vibration module that is capable of two arbitrary frequencies, which is for generation of beat vibration. The overall working principle is clear and the implementation looks robust. The result shows the proposed device can indeed generate the beat vibration, and the signal is strong enough to be sensed by a fingertip. However, I found some flaws in the evaluation, which is potentially due to mistakes during the implementation or measurement. This paper also lacks a review of background literature. I propose a major revision for this paper.
# Main Review
I like the idea of using electrostatic vibration with no moving part. The overall presentation of the implementation is clear, except for a few missing details (see "Minor improvements" section at the bottom). The generated vibration seems to be powerful (at most 2.5g peak to peak) enough to be detected by a human finger.
However, the evaluation result is confusing. The main flaw is presented in Figure 7. Apparently, the device is clearly generating the beat vibration as proposed. In theory, as mentioned as the authors, the frequency of the generated beat vibration should be the difference between two input frequencies. Therefore, on the top row, all 60Hz+65Hz, 90Hz+95Hz, 120Hz+125Hz and 150Hz+155Hz conditions should result in 5Hz beat vibration frequency. However, 60+65Hz and 90+95Hz cases do show 10Hz pattern rather than the expected 5Hz pattern. Other two (120+125Hz, 150+155Hz) conditions seem to exhibit the 5Hz pattern as expected. This fact directly conflicts with the sentence on the paper line #204, "Let us take 90 Hz + 95 Hz as an example where we can easily found two vibration signals (one frequency is 5 Hz and the other is 92.5 Hz)." No discussion about this result is presented in the paper.
There are two possible explanations:
1) ATMega128 microprocessor is an 8-bit AVR controller which does not provide a true DAC (digital-analog converting) functionality. This means whatever analog signal rendered from the microprocessor is the result of PWM (pulse-width modulation, see https://en.wikipedia.org/wiki/Pulse-width_modulation for the idea), which has its own frequency which may affect the result. There are no details about how authors treated it and I couldn't find any filter or DAC related components in the circuit in Figure 3, so I assume authors could not handle this problem properly.
2) Wrong annotation in the plot. For example, the 90+95Hz plot seems denser (=higher frequency) than the 120+125Hz plot. This indicates that the 60+65Hz and 90+95Hz plots are, in fact, plotting 0.0-2.0s time range rather than 0.0-1.0s range.
In the former case (microcontroller fault), authors need to implement a new device that truly generates an analog signal, which requires a major revision. In the latter case of the wrong plot, authors don't need to redo the experiment. But still, authors need to discuss how they generated and treated signals from the microcontroller.
The other main concern is the lack of related works. I was not convinced about the novelty of the method, because the authors don't provide what has been done in the field and the review of related technology. Please be clear that what part of the work is based on existing works and what part is newly introduced.
# Minor improvements
In this section, I am being a little bit nit-picky, but I think it's worth to be considered. Please kindly address those details for the completeness of the paper.
* It's unclear that how the beat vibration allows users to perceived the complex surface of a target object. I can agree that the beat vibration can excite two different mechanoreceptors, but what kind of haptic sensation, in a qualitative manner, could be rendered if the beat vibration is provided? Is there any background work? for example, is there some existing haptic application of beat vibration even with two LRAs(linear resonant actuator)? How an application designer take advantage of having beat vibration?
* Some key details for reproduction of the work are missing. In the setup of Fig 6, what is the shape of the signal generated? (apparently, it's a sine function, but it's not clearly stated). What's the voltage level of the signals? Was the gain of the amplifier also set to 1000? as in Fig 4 setup? How did you generate the sine wave from AVR controller which doesn't have a proper DAC? Could you provide some brief schematic diagram of Fig 3 device?
* What's the rationale behind the choice of 100g weight? Please explain.
* I strongly recommend proofreading by a professional before the publication.
* This is my personal opinion though, as an HCI researcher, I would be happy if authors can provide some application examples with virtual objects as they claimed in the conclusion. It will better convince the HCI community for the need of such a device.
Author Response

(The authors gave the same response as above.)

Round 2
Reviewer 1 Report
The authors proposed electrostatic beat module for tactile stimulation. The paper was improved well from the previous version. However, I think this article still has some flaws to be published in a high-quality journal paper. I would like the authors to clarify the following points;
1. Line 74: The authors claim the necessity of stimulating two or more mechanoreceptors at the same time. Why do we have to use ETB instead of creating signals consisting of waves at multiple frequencies? The authors must claim the advantage of ETB over other approaches. In addition, can ETB stimulate “more” mechanoreceptors? The proposed ETB can create signals at only 2 frequencies.
2. Line 86-94: The authors introduced several prior works. But I cannot see where this work is positioned. Do the prior work have some problems? Does this work solve them?
3. Line 97 to 103: The connection between these two paragraphs does not sound logical. In the former paragraph, the authors claim that the resonant frequency of the actuators in prior work was fixed and can be only used around the frequency. But the proposed ETB also has a fixed resonant frequency and though beat frequency can be changed, the main frequencies cannot be far from the resonant frequency.
4. Line 117: The authors claimed that ETB can simulate a lot of heterogeneous surfaces. How can you say so?
5. Figure 9: Figure 9 shows the raw data. You can show some of them, but you definitely need to show the FFT results to verify the ETB created two vibration frequencies.
6. Experiments: What is the input voltage? Power?
7. Experiments: Is this research approved by the research ethics committee?
8. Conclusions: The authors stated that the proposed actuator could stimulate tow or more human mechanoreceptors. Do you have any evidence?
9. Line 277: These statements are not conclusions.
Author Response

(The authors gave the same response as above.)

Reviewer 2 Report
The article is well modified. However, I cannot perfectly agree with the modified article.
The authors added an experiment. However, the experimet and the result are unnatural. I think that detailed experimental procedure and experimental results should be shown in the article. Please reonsider the experiment.
Also, the autors should confirm that the subjects perceive both low frequency vibration and high frequency vibration to verify the effect of the beat vibration.
I strongly recoend the authors to add a detailed experiment.
Author Response

(The authors gave the same response as above.)

Reviewer 3 Report
Thanks to authors for addressing my comments rigorously. I could see that the paper has addressed most of my concerns. However, still, some more improvement is encouraged.
----------------
The paper needs some more justification regarding a need for the proposed technology. I could convince that the proposed device is working well: it can generate a set of beat vibrations and the generated vibrations are perceivable. However, the claim "beat vibration is distinguishable from pure single-frequency vibrations" is a bit unreliable (see the next part). The paper presents an excellently engineered haptic actuator. This is unquestionably the main engineering contribution.
However, as the paper aims also addressing UI/UX, the paper should be able to answer this question:
"Why such a device is needed?"
Obviously, beat vibration can excite two (or more) different type of mechanoreceptors. However, what is the practical implication of it? The revised paper addressed this question as like:
* line 78: "unlike normal vibration."
* line 261: "the created beat vibration provides different stimulus from pure vibration"
* line 274: "various tactile sensation"
How different it is? Why the beat vibration is better than "pure" vibration? If so, in what way it's better? I believe that there must have been some attempt to use beat vibration in a practical context since it's already known phenomena. Please justify the need for beat vibration. Then the contribution of the paper will be more robust.
----------------
In the experiment (line 247--), Did the participants aware the order of the vibration (starting with beat-vibration, and then pure)? If they already know, and given that the question was just "Yes/No" type, it could be biased. Better ways for testing this are ABX, duo-trio (AXY), and triangle (XXY) tests; participant should be *blind* about the condition.
# For example, in ABX test, the participant only knows A and B are different (but don't know which is the control and which is the test stimulus). X is randomly chosen as A or B. Participants is forced to answer A=X or B=X. In 10 tests, 9+ correct answers are required for 95% confidence level.
Because those tests give a better statistical power, I advise authors to try them. Please search "Discrimination test" for the detailed method.
----------------
Minor comments:
* If you have an early result of haptic application (line 276), could you briefly introduce them, as a practical application example?
Author Response

(The authors gave the same response as above.)

Round 3
Reviewer 1 Report
The authors have made sufficient revision for publication.
I am still not 100% satisfied with the claim "ETB can simulate a lot of heterogeneous surfaces" until the authors show the results that the participants can perceive various kinds of surfaces in the experiments. The correlation between the stimulation and the surfaces is still unclear. I hope the authors discuss this in the next work.
Author Response

(The authors gave the same response as above.)

Reviewer 2 Report
The article is well modified. I will not argue the acceptance.
Author Response

(The authors gave the same response as above.)

Reviewer 3 Report
I thank the authors for addressing most of my concerns. It is now more clear that the beat vibration module is indeed generating a distinctive beat vibration. It'll be more appreciated when the authors can describe how the participants qualitatively feel the beat vibration compared to a pure vibration.
I'd suggest a minor revision to be published. Notably, the writing style could be further improved thorough proofreading.
Author Response

(The authors gave the same response as above.)
